# A Fraction of *Escherichia coli* Bacteria Induces an Increase in the Secretion of Extracellular Vesicle Polydispersity in Macrophages: Possible Involvement of Secreted EVs in the Diagnosis of COVID-19 with Bacterial Coinfections

**DOI:** 10.3390/ijms26083741

**Published:** 2025-04-16

**Authors:** Francisco Sierra-López, Vanessa Iglesias-Vazquez, Lidia Baylon-Pacheco, Emmanuel Ríos-Castro, Juan Carlos Osorio-Trujillo, Anel Lagunes-Guillén, Bibiana Chávez-Munguía, Susana Bernardo Hernández, Gustavo Acosta-Altamirano, Patricia Talamás-Rohana, José Luis Rosales-Encina, Mónica Sierra-Martínez

**Affiliations:** 1Department of Infectomics and Molecular Pathogenesis, Center for Research and Advanced Studies, Av. IPN 2508, Zacatenco, Mexico City 07360, Mexico; luck19861990@gmail.com (F.S.-L.); lbaylon@cinvestav.mx (L.B.-P.); jcosorio@cinvestav.mx (J.C.O.-T.); lagunes402@gmail.com (A.L.-G.); bchavez@cinvestav.mx (B.C.-M.); ptr@cinvestav.mx (P.T.-R.); 2Unidad de Investigación en Salud, Hospital Regional de Alta Especialidad de Ixtapaluca, Servicios de Salud del Instituto Mexicano del Seguro Social para el Bienestar (IMSS-BIENESTAR), Mexico, Carr Mex-Puebla Km 34.5 col., Zoquiapan, Mexico City 56530, Mexico; vanessa.iglesias2711@gmail.com (V.I.-V.); susanamex2007@hotmail.com (S.B.H.); 3Genomics, Proteomics and Metabolomics Core Facility (UGPM) LaNSE, Center for Research and Advanced Studies, Av. IPN 2508, Zacatenco, Mexico City 07360, Mexico; 4Hospital General de México, Eje 2A Sur (Dr. Balmis) No. 148, Cuauhtémoc, Doctores, CDMX, Mexico City 06726, Mexico; mq9903@live.com.mx

**Keywords:** polydispersity EVs, stimulator fraction *E. coli*, SARS-CoV-2

## Abstract

Extracellular vesicles (EVs) can transport molecules that combat viruses, such as RNA against SARS-CoV-2. Bacterial coinfections can help establish certain viruses and worsen diseases. Thus, we designed a model to induce the secretion of polydisperse EVs shown with SARS-CoV-2 and bacterial coinfection using macrophages and *E. coli* fractions as in vitro inducers. We obtained short and large macrophage EVs. The *E. coli* fraction was designated as SDS-soluble bacterial membrane fraction and its associated proteins (SDS-SBMF). The proteins were identified using a mass spectrometer. SDS-SBMF contained mainly OmpF, OmpA, OmpC, OmpX, and lpp. The SDS-SBMF macrophages induced the secretion of polydisperse EVs at 30 min, reaching optimal secretion at 120 min, as observed via scanning electron microscopy and confocal microscopy. Macrophage EVs contained mainly HSP7C, actin, apolipoprotein, GAPDH, annexin A5, PKM, moesin, and cofilin. We observed an increase in EVs in the bloodstream of patients with SARS-CoV-2 and bacterial coinfection, in addition to the presence of SARS-CoV-2 genes (E, ORF) in EVs. This in vitro method for inducing EVs has the potential to be used to obtain larger samples for study and for the detection of diagnostic and prognostic biomarkers of different diseases.

## 1. Introduction

Particularly large groups or populations of extracellular vesicles (EVs) can reach dimensions of up to 10 µm. These are generally polydisperse EVs, or large oncosomes in the case of those released by cancer cells, and have characteristics specific to the cell of origin [1,2,3]. The biomolecules in EVs vary according to the type of cell emitting them and are delimited by a lipid bilayer. This is why accessible compendiums have been generated online, such as EVpedia [4,5], which are used to document and provide consultation for identified biomolecules, such as the proteins most commonly found in EVs. These can include some of the top 100+ EV markers: glyceraldehyde-3-phosphate dehydrogenase (GAPDH), enolase (ENO1), pyruvate kinase 2/3 (PKM2), annexin A2 (ANXA2), L-lactate dehydrogenase A (LDHA), epsilon protein (YWHAE), L-lactate dehydrogenase B (LDHB), profilin 1 (PFN), annexin 5 (ANXA5), cofilin 1 (CFL1), 70 kDa heat shock protein 8 (HSPA8), moesin (MSN), peroxiredoxin 1 (PRDX1), cytoplasmic actin 1 (ACTB), clathrin heavy chain (CLTC), Rab-5c (RAB5C), ezrin (EZR), cyclophilin A (PPIA), and protein 1054 (YWHAB). The proteins that usually adhere to EVs include serum albumin (ALB). Low molecular weight tyrosine phosphatase (LMW-PTP) is not found in most EVs in humans; however, it is found among the top 100+ EV markers identified in colorectal cancer [6], exosomes in urine [7], ovarian cancer exosomes (OVCAR-3 line) [8], pro-coagulant microvesicles (fighting against Streptococcus pyogenes) released by peripheral blood mononuclear cells [9], and thymus exosomes [10], among others.

The cellular machinery used to produce EVs, including the packaging of their cargo and secretion, is commonly manipulated by viruses for their own propagation [4]. For the packaging of viruses in exosomes, the ‘endosomal sorting complexes required for transport’ (ESCRTs) initially intervene, internalizing the cargo in the exosomes within multivesicular bodies [11]. HIV hijacks the machinery for EV formation for its own secretion in structures similar to microvesicles [4,12]. In influenza, viral components have been detected in EVs released by infected cells [13]. Given this, the larger the size of the EVs, the better the capacity to transport viruses and their components, as well as the long chains of nucleic acids. Large EVs can also contain viruses in their interior small vesicles, even those similar to exosomes [14]. Coinfections with viruses and bacteria can worsen diseases; for example, Streptococcus pneumoniae favors influenza, and Escherichia coli worsens infections with rotavirus [15].

The infectious disease COVID-19, caused by the SARS-CoV-2 virus, can cause hyperinflammatory responses in severe cases, where monocytes and macrophages are among the main cells for defense; however, their alteration can be harmful to the host [16]. SARS-CoV-2 can infect multiple organs and cells, including macrophages and monocytes in the lungs, through ACE2-dependent and ACE2-independent pathways, disabling the adaptive immune response [16,17]. Recent studies have shown that SARS-CoV-2 employs small EVs to spread to neighboring cells [18,19].

In patients with SARS-CoV-2, numerous types of antibiotics have been used to prevent possible complications from bacterial coinfections, as viral infections often lead to bacterial pneumonia or secondary bacterial infections, critically increasing the severity of the disease and raising the rate of mortality, as is the case in COVID-19 [20]. The bacterial species most reported in coinfections with SARS-CoV-2 are *Acinetobacter baumannii* (Gram −), *S. pneumoniae* (Gram +), *Pseudomonas aeruginosa* (Gram −), *Staphylococcus aureus* (Gram +), *Escherichia coli* (Gram −) [20,21], *Haemophilus influenzae*, and *Klebsiella* spp. [22].

There are numerous species of bacteria, including Gram-negative bacteria, that have been found in coinfections with different viruses, worsening patient conditions. In the present work, we aimed to design a model to induce the secretion of polydisperse EVs in vitro, representative of the variety of EVs produced in patients with bacterial and SARS-CoV-2 coinfections. For this, we used macrophages and *E. coli* fractions as in vitro inducers of EV secretion, characterized by different techniques. We also analyzed the secretion of EVs in samples from patients diagnosed with COVID-19 and bacterial coinfections, demonstrating the presence of SARS-CoV-2 genes and increases in the surrounding EVs.

## 2. Results

### 2.1. Obtaining a Fraction of Gram-Negative Bacteria E. coli for the Induction of Extracellular Vesicles from Macrophages

The fraction of interest was obtained. It was soluble in a buffer solution containing SDS and was expected to be enriched with components of the bacterial outer membrane. It was designated as the SDS-soluble bacterial membrane fraction and its associated proteins (SDS-SBMF). With 300 mL of *E. coli* culture in LB medium, the optimal yield for our extraction and stocks was prepared with a protein concentration of 0.1 µg/µL for later use as a substrate to purposely stimulate macrophages. The yield was obtained for up to 30 mL of dialyzed SDS-SBMF.

The electrophoretic profile of the SDS-PAGE gels of the SDS-SBMFs showed banding from 6.5 to 200 kD; however, this banding was greater from 31 to 45 kD, mainly an intense band between 39 and 40 kD (Figure 1). The components of SDS-SBMF were identified via mass spectrometry cutting and by analyzing nine consecutive fragments of the SDS-PAGE gel profile (Figure 1, column 6, fragments F1 to F9). In order, the fragments with more banding were F4 > F5 > F3 > F8. The proteins were identified using a MALDI Ekspot and a MALDI-TOF/TOF 4800 plus mass spectrometer (MS) (AB Sciex, Framingham, MA, USA).

The MS/MS spectra were compared using Protein Pilot v. 2.0.1 (AB Sciex, Framingham, MA, USA) against a database of *E. coli* using the Paragon algorithm. The identified SDS-SBMF components of the bacteria are shown in Appendix A, with the most abundant proteins identified in F4, with an outer membrane protein OmpF of 39.3 kD (corresponding to the highest-intensity band) and, subsequently, in F5, with an outer membrane protein OmpA of 37.2 kD; F3, with an ATP synthase beta subunit atpD of 50.3 kD, an ATP synthase alpha subunit atpA of 55.2 kD, an outer membrane protein OmpC of 53.7 kD, a maltoporin lamB of 49.9 kD, a Cytochrome bd-I ubiquinol oxidase subunit 1 cydA of 58.2 kD; and F8, with an outer membrane protein X Ompx of 18.6 kD, a main outer membrane lipoprotein lpp of 8.3 kD, an outer membrane lipoprotein rcsF of 14.1 kD. This suggests that these bacterial fractions mainly correspond to the outer membrane and its associated proteins, which also suggests that they contain LPS corresponding to the outer membrane of *E. coli*.

### 2.2. Induction of EV (Short and Large) Secretion from Macrophages with SDS-SBMF

The SDS-SBMF concentration was standardized to stimulate macrophages into secreting EVs in the visible range of confocal microscopy. The stimulation led the cells to overexpress the low-molecular-weight tyrosine phosphatase (‘LMW-PTP’), also known as acid phosphatase 1 (‘ACP1’), in the cytoplasm and EVs, facilitating the detection of large EVs and short EVs using anti-LMW-PTP polyclonal antibodies (Figure 2). The anti-LMW-PTP antibodies were produced with the recombinant human HsLMW-PTP-b protein. LMW-PTP from mouse (Mus musculus) NP_001103709 (MmLMW-PTP-1) and NP_067305 (MmLMW-PTP-2) have 91% and 89% shared identity with the human GenBank sequence NP_009030 (HsLMW-PTP-b, HsACP1b) (Figure 3D), so the mouse LMW-PTP was recognized and verified as expected.

By standardizing the SDS-SBMF concentration in the cell culture surfaces, we determined a good substrate amount between 10 and 20 ng of SDS-SBMF per cm^2^, using FN to stabilize the substrate. The membrane fraction bacterial SDS-SBMF induced the release of an ‘abundant’ amount of large EVs from the macrophages, with those between 0.5 and 3.0 µm (Figure 2) being very evident. Using DAPI, we verified that the large EVs did not have nuclear fragments (ruling out that they were apoptotic). For the unstimulated macrophages, less than 20% of them were observed with EVs detected with LMW-PTP. Unstimulated macrophages, where EVs were observed in the fluorescence, lacked abundant large EVs (Figure 2, unstimulated). Thirty minutes after induction with SDS-SBMF, macrophages secreting polydisperse EVs were observed. They were also observed at 60 min and 90 min, reaching the optimal time at 120 min.

Using the EVAnalyzer plugin for ImageJ2, we verified the spherical appearance and tendencies of EVs released by the macrophages and an increased area in the Z position of the images in the stimulated samples. This was significant, as shown by the data obtained from the selected representative images (Appendix A). Additionally, we verified that no apoptosis was induced by these SDS-SBMF concentrations with trypan blue. Furthermore, after induction, macrophages showed low but detectable relocalization of LMW-PTP to the nucleus (Figure 2C).

Induced EVs were analyzed using SDS-PAGE, Western blotting (WB), MS, and scanning electron microscopy (SEM). The electrophoretic SDS-PAGE profile of the EVs (using ZnSO_4_, which favors the aggregation of this set of EVs) became more obvious in bands above 40 kD, with apparent protein degradation between 6.5 and 40 kD (Figure 3A). With WB, MmLMW-PTP was recognized using generated anti-LMW-PTP polyclonal antibodies, and a faint band with a molecular weight of approximately 18 kD was observed under denaturing conditions (Figure 3C). An intense band was expected, so the samples were fixed with paraformaldehyde and run for WB. This demonstrated an intense band in the polyclonal anti-LMW-PTP, which remained in the form of a molecular complex above 200 kD (Figure 3B).

Paraformaldehyde intercalates in proteins, preventing their denaturation, and inhibits enzymatic activities, including those of proteases. With paraformaldehyde and cOmplete inhibitors free of EDTA, proteolytic activity in the lysates was largely prevented using WB. Using mass spectrometry, the samples without paraformaldehyde were analyzed, and proteins were identified at molecular weights greater than 40 kD, including serum albumin, protein tyrosine phosphatase receptor type PTPRM 163.6 kD, heat shock cognate protein at 71 kDa, actin at 41.7 kD, and moesin at 67.6 kD (Appendix A). In the F5 fragment, we mainly found isomerase triphosphate at 26.7 kD, apolipoprotein at 30.6 kD, Arhgdia at 23.4 kD, GAPDH at 38.6 kD, annexin A5 at 35.7, galectin at 27.4 kD, hemoglobin at 15.8 kD, and cofilin at 18.7 kD. The rest of the proteins, identified as Unused >1.3, are listed in Appendix A.

Macrophages cultured on the bacterial fraction of *E. coli* as substrates (SDS-SBMFs) were analyzed via scanning electron microscopy (SEM). They showed different states of activation in EV secretion because they did not do so in a synchronized manner, making it possible to find macrophages releasing EVs similar to exosomes (Figure 4B). In greater quantities, macrophages with large EVs on their surfaces were observed after 2 h of stimulation, which were released in the form of ‘clusters’, as seen on the exterior of Figure 4C. Macrophages also released large EVs individually (Figure 4D). The production of large EVs exceeding 0.5 µm and even 1 µm was confirmed via SEM. In the samples that were fixed with large EVs on their surface and subsequently subjected to the extraction of those EVs with SDS-Triton X100 detergents, it was possible to observe cavity-like indentations where the large EVs had been, suggesting that there is a limiting lipid membrane or segmentation between these large EVs and the originating emitting cell before it finishes releasing them (Figure 4, Ext.).

The viability of the samples was checked, and 95% of the cells survived at 2 h. The number of large EVs (greater than 500 nm) that were secreted or remained associated with the external membrane of the macrophages was determined by directly observing the cultures via optical microscopy, confocal microscopy, and SEM. Subsequently, after 2 h of inductive culturing on the SDS-SBMF surfaces, we estimated that the number of large EVs varied between 40 and 80 for each cell that showed them. When performing the immunodetection method for confocal microscopy, numerous LMW-PTP-positive large EVs detached from the surface of the cells, leaving approximately 15–40 large EVs on the surface of the macrophage (Figure 2 and Figure 4). Figure 4 shows representative images of some of the combinations of structures, starting with the reactions of unstimulated macrophages to stimuli.

### 2.3. Bacterial Co-Infection Analysis in COVID-19 Patients

Starting with the premise that in the event of an infectious process, the secretion of a wide variety of EVs increases, we correlated the increase in these vesicles by associating them with the severity of the infection, in this case, COVID-19. Therefore, 83 samples from patients diagnosed with COVID-19 were analyzed via qRT-PCR. The patients were selected according to the following inclusion criteria: a mean age of 57.20 years, with a range of 18 to 60. In total, 33.73% of the patient population were women, and 66.26% were men. All patients had moderate to severe disease according to WHO criteria, with an average oxygen saturation of 69% (all patients were hospitalized). Additionally, the presence of bacterial coinfections with the following species was analyzed: *Streptococcus pneumoniae*, *Staphylococcus aureus*, *Klebsiella pneumoniae*, *Pseudomonas aeruginosa*, and *Acinetobacter baumannii*. The results showed that 98.8% of the patients with COVID-19 presented with one or more bacteria (Appendix A). Only 1.2% of the patients did not present with bacterial coinfections. In total, 90.4% of patients presented with two or more species of bacteria of interest in coinfection with COVID-19. The most recurrent species in multiple coinfections were *Staphylococcus aureus* and *Streptococcus pneumoniae* in 45.8% of the patients. Gram-negative *Klebsiella pneumoniae* was present in 39.7% of patients with multiple coinfections. *Acinetobacter baumannii* was present in only *n* = 3 patients because the samples were taken at the beginning of the hospital stay. In our study, 85.54% of the patients died.

### 2.4. Analysis of EV Size and Concentration in Serum Applied to Patients with COVID-19

EVs were obtained traditionally using serial centrifugation and ultracentrifugation of up to 110,000× *g*, using plasma obtained from the peripheral blood of patients diagnosed with COVID-19. The results from a Nano Sight (NS) assay showed a trend demonstrating more particles (EVs) in patients infected with COVID-19 and coinfected with bacteria compared with healthy patients without COVID-19 (Figure 5). The patients analyzed were selected randomly to be representative of the overall sample. Figure 5 shows that triplicates of the particle/EV detections from the patients with COVID-19 were similar between them, mainly demonstrating an increase in particles related to EV populations between 100 and 200 nm. The traditional strategy of obtaining particles/EVs via ultracentrifugation appeared to be ineffective for obtaining EVs with dimensions larger than 200 nm, resulting in a low number of particles in these ranges. They were sporadically detected as particles of larger dimensions than those close to 650 nm, without being abundant.

### 2.5. Detection of SARS-CoV-2 in EVs via PCR

After RNA extraction from the extracellular vesicles, SARS-CoV-2 was detected via dPCR. Figure 6 shows that the virus was detected in the EVs of patients with mild, moderate, or severe clinical manifestations, demonstrating 0.2–0.5 copies/µL in each group. Healthy donors did not show the viral material of interest. The results show that EVs increased in patients infected with COVID-19 and coinfected with the bacteria analyzed, suggesting that these EVs may be used to transport viral material (Figure 6).

## 3. Discussion

Reports over the last decade have shown that macrophages can release ‘extracellular vesicles’ (EVs) in a spontaneous, unsynchronized manner. Of these macrophages, the J774A.1 line is usually used as a model for studying this cell type, as it secretes EVs with sizes ranging from 50 to 200 nm, with a higher frequency of those close to 130 nm and a lower frequency of those between 200 to 300 nm, according to ‘nanoparticle tracking analysis’ [23]. In the presence of Gram-negative bacteria, such as Mycobacterium tuberculosis, the size of particles/EVs emitted by macrophages slightly increases (150 nm EVs), along with the appearance of populations close to 350 nm at a low frequency [24]. This increases the release of small EVs similar to exosomes when these cells (or others participating in the immune system) interact with components of the outer membrane of bacteria, such as lipopolysaccharides (LPS). This has also occurred in study models of alveoli [24,25]. Given the above, we used mass spectrometry (MS) to analyze the fraction of Gram-negative Escherichia coli that we extracted. The MS results confirm that the proteins obtained in greater abundance in the *E. coli* fraction used to stimulate macrophages belong to the outer membrane of the bacteria.

We mainly focused on identifying the protein content of the fraction that we called SDS-soluble bacterial membrane fractions and their associated proteins (SDS-SBMFs). The most abundant protein was the outer membrane protein F OmpF. The next most abundant proteins were OmpA, ATP synthase beta subunit atpD, ATP synthase alpha subunit atpA, maltoporin lamB, cytochrome bd-I ubiquinol oxidase subunit 1 cydA, outer membrane protein X Ompx, main outer membrane lipoprotein lpp, and outer membrane lipoprotein rcsF. Given the methodological strategy we used to obtain SDS-SBMFs, this suggests that they should have the corresponding lipopolysaccharide (LPS) content proportional to the amount of membrane obtained, although we did not analyze this content.

Various EV populations are secreted by a wide variety of organisms and eukaryotic cells, as this is an evolutionarily ancient conserved mechanism [26]. They have functions such as exchanging cellular components and information exchange; have immunomodulatory properties; and can transport various biomolecules, such as lipids, sugars, and proteins (enzymes) [27,28,29,30], and nucleic acids such as RNA and DNA [26,31].

Nonetheless, the cellular machinery used to produce EVs is commonly manipulated by viruses to propagate themselves in the cells they infect [4]. This has been observed in the hepatitis A virus (HAV) in exosomes with multivesicular bodies [11] and in HIV in structures similar to microvesicles [4,12] in influenza [13]. It was also recently documented that the SARS-CoV-2 virus uses small EVs to spread to neighboring cells [18,19].

Our results show the E and ORF genes of SARS-CoV-2 in the EV content, suggesting that some EV populations can spread infections to other cells. SARS-CoV-2 has even been shown to be capable of infecting macrophages and monocytes in the lungs through ACE-2-dependent and ACE-2-independent pathways, disabling the adaptive immune response of the main cells involved in the defense carried out by the immune system [16,17]. This demonstrates the importance of macrophage EVs in SARS-CoV-2 infections, as these cells are considered among the main sources of EVs in fluid and lung tissues.

We propose a symbolic scheme that shows the dynamics of EV secretion in bacterial coinfections with SARS-CoV-2, where the virus takes advantage of the generation of many EV populations for its propagation (Figure 7).

We chose this type of cell to obtain a range of EVs in vitro that may be present in patients in the context of bacterial coinfection. Viral and bacterial coinfections can worsen a disease. This often occurs with *Streptococcus pneumoniae*, which promotes influenza, and infections with *Escherichia coli*, which promotes rotavirus infections [15]. Taken together, we have explored the possibility that bacterial components favor the production of EVs. Therefore, our working group decided to include patients infected with SARS-CoV-2 in this study, showing that a large percentage of these patients had a bacterial infection. Increased extracellular vesicles in COVID-19 coinfections could be a biomarker correlated with disease severity. Notably, COVID-19 patients who develop a bacterial coinfection are generally admitted to the intensive care unit (ICU) and have a higher risk of death.

Using the SDS-SBMF fraction of *E. coli*, we managed to induce a notable increase in the production and secretion of large EVs from macrophages. In general, the vast majority of EV counts were represented by small exosome-like EVs; these could foreshadow a significant increase in the production of large EVs, a strategy commonly used in particle counts employing Nano Sight, as has been reported regarding particle/EV counts for macrophages [23,24]. In this case, the variety of EVs is demonstrated by a histogram, given that a particle/EV close to 1000 nm has a much greater content capacity and variety of cargo than a particle/EV of 100 nm. In this analysis, the immense increase in small EVs could overshadow the increase in large EVs. Our results suggest that histograms should be considered separately for EVs that are less than 200 nm to avoid masking possible increases in the secretion of large EVs. The results also suggest that the strategy we used to obtain the variety of small to large EVs—as observed using SEM and SDS-PAGE—could be considered for future work seeking to better adapt the strategy to mass spectrometry and other analyses. We were able to monitor the EVs we induced through detection with antibodies against the low-molecular-weight protein tyrosine phosphatase (LMW-PTP), which is not common in the vast majority of EVs released by most cells in the human body. This increases specificity when detecting EVs emitted by macrophages. Cells that emit EVs transporting LMW-PTP include colorectal cancer [6], exosomes in urine [7], exosomes from ovarian cancer (OVCAR-3 line) [8], pro-coagulant microvesicles released by peripheral blood mononuclear cells (combatting Streptococcus pyogenes) [9], and thymus exosomes [10].

The large EVs that we induced with the *E. coli* fraction were easily observable via light (40×) and confocal microscopy. They were abundant, generally exceeding 500 nm and often exceeding 1000 nm. The large EVs left cavity-like indentations when mechanically removed from the surface of the fixed macrophage, as shown using SEM. These data, along with an analysis of images obtained using confocal microscopy, suggest that these EVs are rich in cytoplasmic and vesicular content, including LMW-PTP. The proteins identified using MS in the EVs were serum albumin, 71 kDa heat shock cognate protein, radixin, ezrin, actin, moesin, triosephosphate isomerase, apolipoprotein, Arhgdia, GAPDH, annexin A5, and cofilin. Albumin is frequently found in EVs obtained in vitro and in vivo, suggesting that they are a frequent contaminant or probable stabilizer of EVs [33]. In our case, this was interesting, as prior to induction, the serum was removed; thus, the cells likely provided albumin or phagocytosed fragments of this protein that were partially redirected to the EVs. The previously identified components of EVs secreted by macrophages using the J774A.1 line include actin, moesin, GAPDH, radixin, ezrin, and annexin A5 [23,34], suggesting that we obtained these EVs from macrophages with suitable specificity. In the EVs of human monocytes, cofilin, apolipoproteins, and even histone H2A have been reported [35]. Arhgdia (Q99PT1 UniProt access) has been identified in exosomes secreted by fibroblasts in an environment that promotes the migration of breast cancer cells [36].

In confocal fixed-sample assays, LMW-PTP was shown to be intense and easy to detect; in EV extracts, it showed strong degradation in Western blot (WB). In WB, degradation was avoided by using paraformaldehyde on the samples; however, this caused LMW-PTP to remain part of high-molecular-weight molecular complexes greater than 200 kD, and a denatured without a fixative appeared at approximately 18 kD (expected weight).

Our working group will evaluate applications in different cell types, simulating the secretion of polydisperse EVs to analyze diseases. The method we have generated provides a relatively simple way to produce abundant amounts of macrophage EVs in a short time, requiring common equipment in laboratories where cell cultures are performed.

## 4. Materials and Methods

### 4.1. Human Samples for Assays

Male and female patients aged 18 to 60 years with suspected SARS-CoV-2 infection were selected. All samples were taken at the beginning of the hospital stay. The participants agreed (controls and patients with mild infection) to participate in the study and signed informed consent based on complete and exhaustive information. Patients with HIV, pregnant women, oncohematological patients, and patients with any autoimmune conditions were excluded from the study. On the first day of admission for hospitalized patients (moderate and severe), family members and legal guardians gave consent for the use of patient data and samples (swabs and plasma) for research activities by signing digital documents that remain in the custody of the SALUDNESS platform. The study was approved by the Ethics Committee of the Hospital Regional de Alta Especialidad de Ixtapaluca, “IMSS-Bienestar” (NR-074-2023). The procedure for obtaining peripheral blood from volunteers was carried out under the international guidelines established for the study of human populations by the Declaration of Helsinki.

### 4.2. Cell Culture

J774A.1 murine macrophage cell lines from BALB/c mice were cultured in high-glucose-supplemented DMEM (Dulbecco’s Modified Eagle Medium) with 10% (*v*/*v*) previously inactivated fetal bovine serum (FBS) (PAA, A15-701), 100 IU/mL of penicillin, and 100 µg/mL of streptomycin. The cells were incubated at 37 °C in a humidified incubator with 5% CO_2_ and 95% air. The cells were FBS-starved prior to EV inductions and assays.

### 4.3. Cloning, Expression, and Purification of rHsLMW-PTP

Total RNA was obtained from 5 × 10*6* HeLa cells using the TRIzol method (Invitrogen Life Technologies, Grand Island, NY, USA). The retro-transcription (RT) and PCR were standardized (Appendix A).

The coding sequence for the HsLMW-PTP gene (GenBank NM_007099.3) encoding protein NP_009030.1 (HsLMW-PTP isoform b, ACP1b) was obtained via PCR using the following primers: 5′-CAGGAATTCTCAGTGGGCCTTCTCCAAGAACGC-3′ (antisense) with EcoR1 restriction site (underline), and 5′-CGGATCCATGGCGGAACAGGCTACCAAGTC-3′ (sense) with BamH1 restriction site (underline). The genes were subcloned using the pCR^®^4-TOPO plasmid and cloned in the pRSET-A expression vector. The recombinant histidine-tagged protein HsLMW-PTP was obtained as described by Sierra-López, 2021 [37]. The nucleotide sequence of HsLMW-PTP was corroborated using the Perkin Elmer/Applied Biosystems 377-18EInDNA Sequencer (Waltham, MA, USA) from the Department of Genetics and Molecular Biology of Cinvestav, Mexico.

### 4.4. Mice Immunization Protocol and Obtaining Anti-HsLMW-PTP Serum

Four female BALB/c mice (6–8 weeks old) from CICUAL (Cinvestav, Mexico) were used according to institutional animal care guidelines. The mice were immunized via intraperitoneal (i.p.) injection with 10 µg of rHsLMW-PTP emulsified in Freund’s complete adjuvant (FCA) and boosted twice with 10 µg of the recombinant protein in incomplete Freund’s adjuvant (IFA) (Sigma, St. Louis, MO, USA) every 2 weeks. Then, the animals were bled to obtain anti-HsLMW-PTP immune serum.

### 4.5. Extraction of Escherichia coli Fraction

Gram-negative *E. coli* strain BL21(DE3)pLysS bacteria was incubated in 300 mL of Luria–Bertani (LB) medium overnight at 37 °C with orbital shaking. The bacteria were pelleted for 10 min at 3500× *g* in a JA10 rotor. The pellet was resuspended in 40 mL of a buffer solution, pH 8.0, with 10 mM Tris-HCl, and lysed via sonication (3 pulses of 60 s, 90% amplitude) in cold conditions. It was then centrifuged at 7000× *g* for 5 min (Eppendorf Centrifuge 5415C, Sigma-Aldrich, Sait Louis, MO, USA) in 1.5 mL conical tubes. The supernatant was recovered (avoiding the pellet) and centrifuged at 11,800× *g* for 30 min. The sediment was recovered, combined, and resuspended in 20 mL of buffer solution with 10 mM Tris-HCl pH 8.0, MgCl_2_ 10 mM, and 2% Triton X-100. The samples were vigorously shaken for 2 min and incubated for 30 min at 37 °C. They were then centrifuged at 14,000× *g* for 30 min. The supernatant of the Triton-soluble components was collected (T-SBMF). The sediment was vigorously resuspended for 2 min with 15 mL of a 10 mM Tris-HCl pH 8.0 solution, 10 mM MgCl_2_, and 2% SDS and collected (SDS-SBMF). In a cold room (temperature below 15 °C), separately, the samples in the buffer solution with SDS and the buffer solution with Triton X-100 were dialyzed with constant stirring using a 15,000 Dalton membrane with 4 L of milliQ water per sample, ending the dialysis after 72 h. We obtained 30 mL of dialyzed SDS-SBMF. The method was modified from Zepeda Cresto, Vidal Vilches, and Sáenz Iturriaga, 2012 [38]. Protein concentration was quantified using the Lowry method.

### 4.6. Standardization of Macrophage EV (Short and Large) Secretion and Isolation

Glass slides or culture surfaces were coated with a mix of 0.250 µg of FN and 10–20 ng of dialyzed SDS-SBMF protein per cm^2^ (stock: 200 µg of FN and 10–20 µg of SDS-SBMF by mL). They were quickly dried in air in a laminar flow hood and sterilized under UV light for 10 min. FN was obtained as indicated by Sierra-López, 2018 [39] and used to enhance the stable dispersion of the SDS-SBMF substrate (dialyzed) used as a stimulant for EV secretion. We placed 4 × 10 J774A.1-line macrophages in DMEM (without serum) per cm^2^ of the culture surface with or without a stimulant (substrate). The cells were monitored under an inverted microscope. We fixed the samples at 30 and 120 min of stimulation with 4% paraformaldehyde in PBS for 45 min, taking care not to mechanically detach the EVs for confocal microscopy assays. We chose to recover the EVs from the supernatants at 120 min.

The macrophage culture supernatants were incubated for 120 min on an FN-SDS-SBMF substrate in Petri dishes (diameter 10 cm) with 15 mL of DMEM (without serum). They were first suspended via orbital movements, and 14 mL were collected. The supernatants were distributed in 1.5 mL conical tubes (Eppendorf) and centrifuged at 120× *g* for 5 min to sediment the cells. The supernatants were recovered by discarding the pellet together with 100 µL of adjacent supernatants. In total, 0.2 mM ZnSO_4_ was added to the supernatants, and the EVs were pelleted via centrifugation at 12,000× *g* for 20 min (Centrifuge 5415C).

### 4.7. SDS-PAGE and Western Blotting

EV extracts and SDS-SBMF proteins were resolved on 12% SDS-PAGE [40] and either visualized via staining with Coomassie blue or electrophoretically transferred to a nitrocellulose membrane at 80 V. In total, 8 µL of β-mercaptoethanol and SDS-PAGE loading buffer was added to the samples, which were immediately placed in boiling water for 8 min. The EV samples showed degradation of LMW-PTP detection, for which some samples were treated with cOmplete protease inhibitors and paraformaldehyde, boiled for 8 min, and cooled on ice, followed by adding 8 µL of β-mercaptoethanol, as indicated by Sierra-López et al., 2021 [37]. Nitrocellulose Western blots (WBs) were incubated with mouse anti-LMW-PTP polyclonal antibody at 1:1000 dilution. Bound antibodies were detected using alkaline phosphatase-conjugated goat anti-mouse IgG (H + L) (Invitrogen, Camarillo, CA, USA) at 1:4000 dilution in TBST. WBs were then developed with an NBT/BCIP substrate kit (Invitrogen, Camarillo, CA, USA).

### 4.8. Confocal Microscopy

Macrophages were unstimulated or stimulated for 30 to 120 min with FN-SDS-SBMF on coverslips fixed with 4% p-formaldehyde in PBS for 45 min at 37 °C; then, they were washed with PBS. Permeabilized was standardized at 0.0015% SDS and 0.0006% Triton X-100 in PBS for 8 min at room temperature (high detergent concentrations, such as 0.4% SDS and 0.2% Triton X-100, destroy large and short EVs, and 0.01% SDS with 0.004% Triton X-100 decouple EVs) and washed carefully with PBS once. The samples were blocked with 5% bovine serum and washed. The cells and EVs were incubated with either serum anti-LMW-PTPs (1:300) in PBS for 1 h, followed by anti-mouse FITC-conjugated secondary antibodies (Zymed 1:300), or Rhodamine–Phalloidine (Sigma; 1:150) for 1 h at room temperature. Samples were preserved using Vectashield with DAPI Antifade Reagent (Vector, Newark, CA, USA), examined through a Carl Zeiss LMS 700 confocal microscope, and processed with ZEN 2010 (Zeiss, White Plains, NY, USA). The color green was used to represent LMW-PTP to analyze EVs with the ImageJ2 (version 2.16.0) EVAnalyzer plugin (ImageJ/Fiji).

### 4.9. Scanning Electron Microscopy (SEM)

Macrophage–EV samples were fixed with paraformaldehyde after being permeabilized with 0.0015% SDS and 0.0006% Triton X-100 or 0.01% SDS and 0.004% Triton X-100 to decouple EVs. The samples were washed with PBS, again fixed with 2.5% (*v*/*v*) glutaraldehyde in 0.1 M sodium cacodylate buffer, pH 7.2, and dehydrated with increasing concentrations of ethanol. The samples were then critically point-dried with CO_2_ in a Samdri-780 Tousimis apparatus. Afterward, in an ion-sputter device (Jeol-JFC-1100), they were gold-coated and examined with a Jeol JSM-7100F field emission scanning electron microscope [41].

### 4.10. Liquid Chromatography and MALDI-MS/MS

EV extracts without p-formaldehyde from stimulated macrophages or bacteria fragment SDS-SBMF were resolved in 12% SDS-PAGE and stained with Coomassie blue, as described above. SDS-PAGE gel fragments between 10 and 200 kDa were worked on by the Genomics, Proteomics, and Metabolomic Service of LaNSE (CINVESTAV IPN) according to the modified Shevchenko protocol [42] and the modified method by Barrera-Rojas et al., 2018 [43]. The resulting MS/MS spectra were compared using Protein Pilot v.2.0.1 (ABSciex, Framingham, MA, USA) and the Paragon algorithm [44] against *Escherichia coli* K12 (Uniprot, 4309 protein sequences database) or Mus musculus (Uniprot, 46,452 protein sequences database). The detection threshold was 1.3 to ensure 95% confidence. The identified proteins were grouped using the ProGroup algorithm to minimize redundancy.

### 4.11. Detection of SARS-CoV-2 via qRT-PCR from Nasopharyngeal Swabs

RNA was extracted from nasopharyngeal samples following the protocol recommended by FAST GENE for the RNA Virus Kit (Cat. No. FG-82300). The RNA was eluted by adding 50 µL of nuclease-free water previously heated to 60 °C. The RNA contained in the extracts was quantified through absorbance using Epoch™ equipment (Bio Tek, Charlotte, VT, USA).

The SARS-CoV-2 virus was detected via qRT-PCR using specific primers and probes for the E and ORF genes. In total, 250 ng of RNA extract was used and subjected to qRT-PCR using the Luna Universal Probe One-Step RT-qPCR Kit (Cat. No. E3006L). This experiment was performed in multiplex. E gene: Fw-ACAGGTACGTTAATAGTTAATAGCGT, Rv-ATATTGCAGCAGTACGCACACA, Probe: ACACTAGCCATCCTTACTGCG, Tm 55 °C. ORF1ab gene: Fw-CCCTGTGGGTTTTACACTTAA, Rv-ACGATTGTGCATCAGCTGA, Probe CCGTCTGCGGTATGTGGAAAGGTTATGG, Tm 56 °C.

### 4.12. Analysis of Bacterial Coinfection in Patients with COVID-19 via Real-Time PCR

Blood, nasopharyngeal swabs, oropharyngeal swabs, and bronchial lavage samples were collected from April 2020 to September 2021 at the Ixtapaluca Regional High Specialty Hospital in the COVID-19 emergency unit and the intensive care unit from 100 patients with symptoms associated with COVID-19. This project was evaluated by research ethics and biosafety committees and authorized under no. NR-074-2023.

The PCR test was performed, and the diagnosis was based on reagents approved by the Institute of Epidemiological Diagnosis and Reference (InDRE). Pathogen detection was directed at *Acinetobacter baumannii*, *Klebsiella pneumoniae*, *Pseudomonas aeruginosa*, *Staphylococcus aureus*, and *Streptococcus pneumoniae* using specific primers and probes (Appendix A).

### 4.13. Traditional Ultracentrifugation Method and Nano Sight Analysis of EV Size and Concentration in Serum Applied to Patients with COVID-19

To isolate EVs, 300 μL of plasma and 900 μL of sterile 1X PBS underwent serial centrifugations at 4 °C. The samples were ultracentrifuged at 110,000× *g* at 4 °C with a 3-micron vacuum for 1 h 30 min using a Rotor 70 Ti (Beckman, Brea, CA, USA) in a SovallTM WX centrifuge. Finally, 200 µL of TRIzol Reagent from Invitrogen (Cat. No. 15596026) was added. The samples were stored at −80 °C until use.

For the Nano Sight assay, intact EV samples were diluted in 1 mL of 1X PBS. All adjustments required by the equipment were followed with respect to the NanoSight NS300 user manual (Malvern Instruments Ltd., Malvern, UK) using a 488 nm laser. The detection threshold of the camera was increased until all the particles were visible. Three 60 s videos were documented for each sample. The data were then analyzed with NTA 3.0 (Malvern Instruments Ltd.). Triplicates for COVID-19 patients and health donors were performed.

### 4.14. Detection of SARS-CoV-2 in EVs Using Digital PCR

After RNA extraction from the EVs, the RNA was subjected to a reverse transcription reaction using the LunaScrip RT SuperMix Kit (Cat No. E3010L). Subsequently, we carried out a PCR reaction using the digital dPCR technique with the commercial kit QIAcuity EG PCR (Cat No. 250113) based on the supplier’s specifications. The oligonucleotides used for the ORF1ab gene were as follows: Fw-CCCTGTGGGTTTTACACTTAA, Rw-ACGATTGTGCATCAGCTGA. The nanoplates were subjected to dPCR incubation at 95 °C for two minutes, with 40 cycles at 95 °C for 15 s and 60 °C for 30 s. Additionally, the plates were subjected to 500 units of exposure time and a gain of 6.

### 4.15. Statistical Analysis

Confocal analysis of EV secretion by macrophages was performed using Fiji and the EV Analyzer plugin (version 8.1.3 beta). Function: EV count; identical thresholds: 10; threshold method: ‘Li’; min circularity: 0.5; filter type: EV-GFP (to FITC channel); particle size range: 1–999,999. In unstimulated macrophages, cells that secrete EVs were sought and analyzed (<20% cells). The EVAnalyzer data description was analyzed using GraphPad Prism 5, one-way ANOVA analysis of variance, and Bonferroni’s multiple comparison post-test.

## 5. Conclusions

We have developed a method to induce macrophages to secrete a wide variety of polydisperse populations of extracellular vesicles in vitro through the bacterial fraction of *E. coli*. A variety of populations could be present in bacterial coinfections with SARS-CoV-2, which we believe has participated in aggravating clinical manifestations in patients, increasing the spread of the virus, as 85.5% of the patients in this study died. Our in vitro method of inducing extracellular vesicles could be used to obtain larger samples for study and the detection of diagnostic and prognostic biomarkers of different diseases.

## Figures and Tables

**Figure 1 ijms-26-03741-f001:**
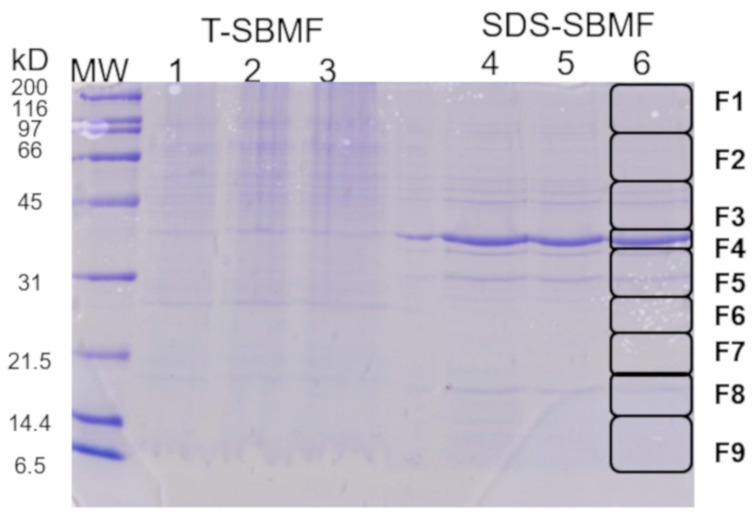
SDS-PAGE profiles of *E. coli* bacteria fraction used to stimulate macrophage EV secretion. Triton-soluble bacterial membrane fractions (T-SBMFs) (1, 2, 3) and SDS-soluble bacterial membrane fractions and their associated proteins (SDS-SBMF) (4, 5, 6) were obtained from *E. coli* BL21 (DE3) pLysS. The SDS-SBMF protein profile was cut into 9 fragments for mass spectrometry. MW: Molecular Weight Standard, Broad Range (Bio-Rad, Hercules, CA, USA).

**Figure 2 ijms-26-03741-f002:**
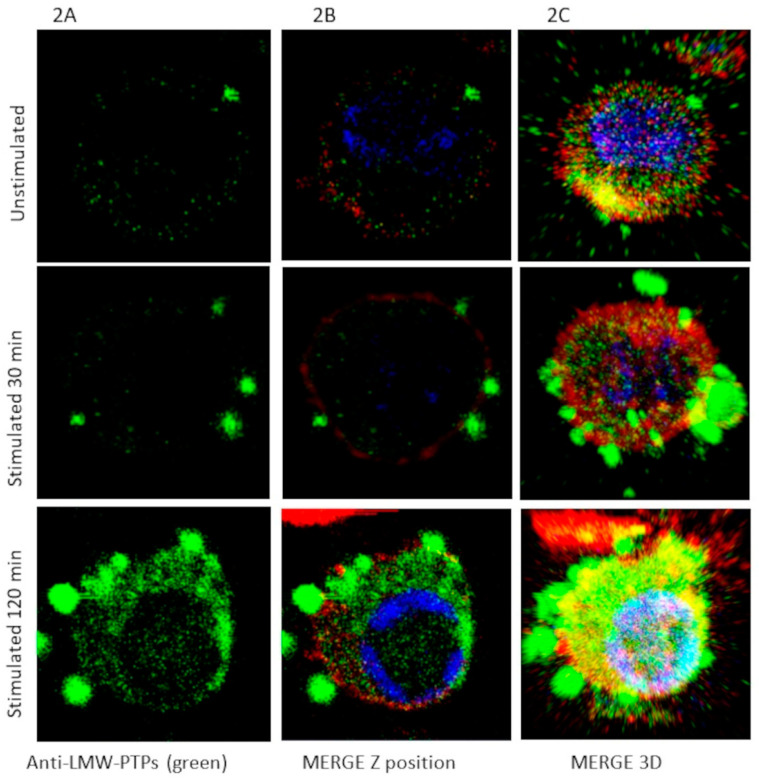
Large and short EVs were induced by the bacterial fraction of *E. coli* and recognized by anti-LMW-PTPs. Macrophages were stimulated for 30 and 120 min at 37 °C on bacterial extract (SDS-SBMF), or unstimulated (cells unstimulated that released EVs were <20%). Mouse anti-LMW-PTPs (**A**–**C**) recognized by anti-mouse IgG FITC (green), phalloidin-rhodamine (F-actin, red), and DAPI were used. Images are representative of 10 assays. Each image is 20 × 20 µm.

**Figure 3 ijms-26-03741-f003:**
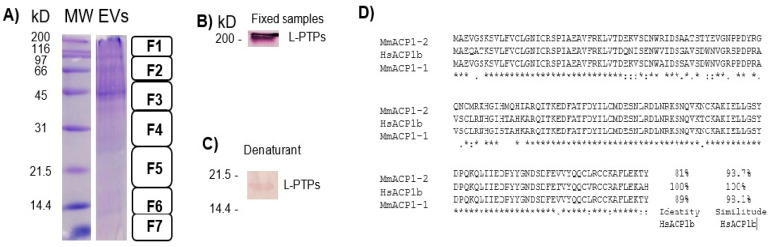
SDS-PAGE and Western blot examination of EVs isolated from macrophages stimulated with SDS-soluble bacterial membrane fractions and their associated proteins (SDS-SBMF). (**A**) Short and large EVs mixed secreted by approximately 1 × 10^5^ macrophages were precipitated using ZnSO_4_, and resolved in SDS-PAGE. Sections F1–F7 were cut using mass spectrometry analysis. By WB, LMW-PTPs were recognized by anti-LMW-PTPs in fixed boiled ((**B**)—18 kD) or denaturant (**C**) samples. The protein was mostly recognized in fixed samples, where they remained in complexes larger than 200 kD. MW: Molecular Weight Standard, Broad Range. Representative image of 3 independent assays. Aligment of MmACP1-1, MmACP1-2, and HsLMW-PTP-b (HsACP1b) by Clustal O showed highly conserved sequences (**D**). The recombinant protein HsLMW-PTP-b was used to generate anti-LMW-PTP polyclonal antibodies.

**Figure 4 ijms-26-03741-f004:**
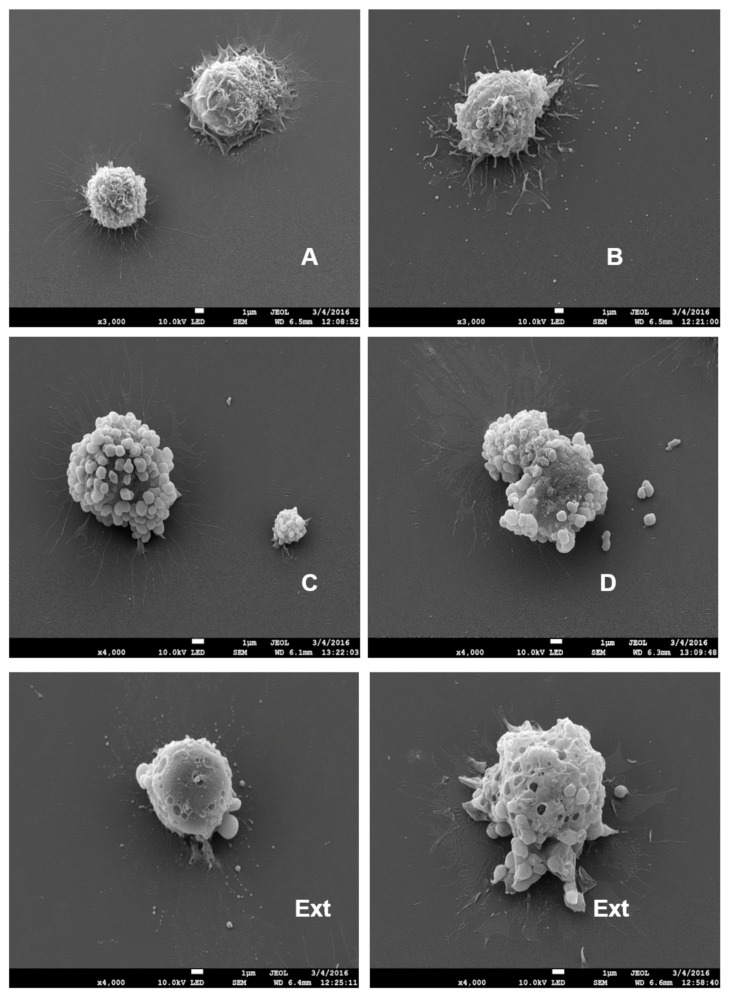
Scanning electron microscopy (SEM) of macrophages with EVs. Representative images of 3 assays. (**A**–**D**) shows the sequential process of bacteria-fraction (SDS-SBMF) stimulation to the release of macrophage EV polydispersity in sizes. Ext: Macrophages that were fixed with EVs anchored to their surface (similar to (**C**)) were subsequently depleted of EVs using detergents, and then SEM was performed, showing numerous indentations on the cell surface.

**Figure 5 ijms-26-03741-f005:**
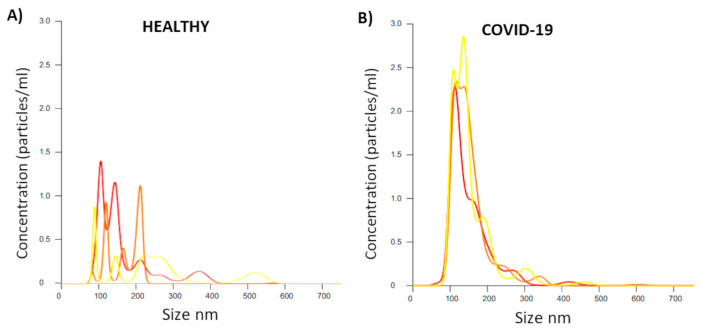
Size of EVs obtained using ultracentrifugation from the plasma of healthy individuals and patients diagnosed with COVID-19. EVs in plasma from healthy (**A**) and COVID-19-diagnosed patients (**B**) were analyzed on NanoSight NS300 equipment. Each color (red, orange, and yellow) corresponds to technical repetitions.

**Figure 6 ijms-26-03741-f006:**
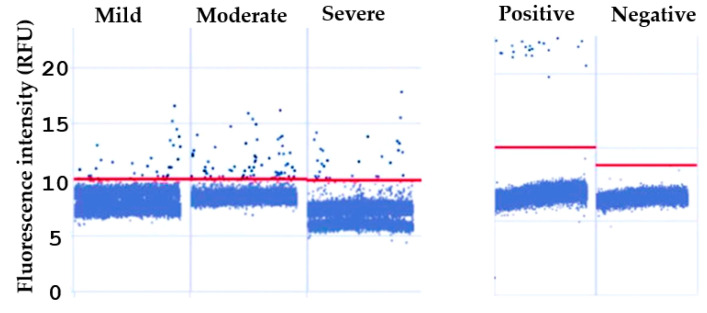
Detection of SARS-CoV-2 in EVs via dPCR. The SARS-CoV-2 virus was found in EVs via PCR in patients diagnosed with COVID-19 with mild, moderate, or severe clinical manifestations. For each group, 0.2–0.5 copies/µL were found. No viral material was detected in the healthy subjects.

**Figure 7 ijms-26-03741-f007:**
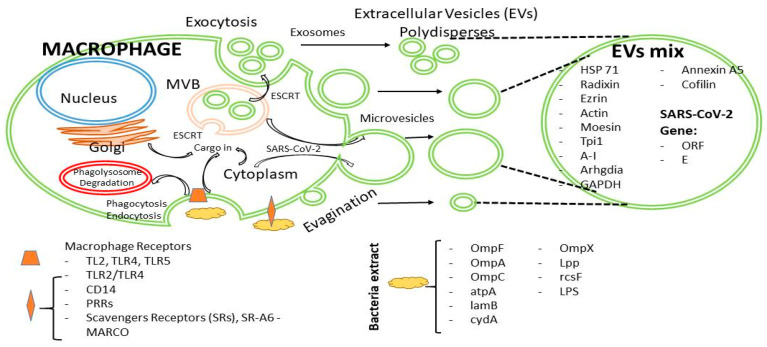
Schematic model of polydisperse extracellular vesicle secretion by macrophages in the context of bacterial-SARS-CoV-2 coinfection. Initially, the cytoplasmic membrane receptors of macrophages come into contact with the components of the bacterial outer membrane. Toll-like receptors (TLRs), pattern recognition receptors (PRRs), scavenger receptors, CD14, and other receptors are involved [32]. Among the bacterial outer membrane ligands that could act synergistically in the stimulation of macrophages are the ‘Outer Membrane Proteins’ (OmpF, OmpA, OmpC, OmpX), atpA, lamB, cydA, lpp, rcsF, and LPS. Ligand–receptor interactions that trigger cytoskeletal remodeling processes involved in phagocytosis, endocytosis, de novo protein synthesis, and extracellular vesicle (EV) production. EVs are produced by two general mechanisms, one involving the formation of multivesicular bodies (MVBs) containing exosomes, and the second involving evagination or budding outwards from the plasma membrane, culminating in the release of microvesicles. The formation of different or polydisperse EVs involves the recruitment of different components that can come from the Golgi apparatus, cytoplasm, plasma membrane, among other compartments or organelles, and the participation of the ‘endosomal sorting complex required for transport’ (ESCRT) machinery. Cells that are infected with SARS-CoV2 or some other viruses may release EVs with ‘virus genes’ as cargo. Macrophages release populations of EVs as exosomes and short and large microvesicles. Large EVs have a higher cargo capacity and, therefore, larger components than small EVs.

## Data Availability

https://www.mdpi.com/2076-2607/11/11/2762, https://www.frontiersin.org/journals/cellular-and-infection-microbiology/articles/10.3389/fcimb.2018.00295/full, https://www.sciencedirect.com/science/article/abs/pii/S030090842030273X?via%3Dihub, (accessed on 3 April 2025).

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
