# Peer review of "A Fraction of Escherichia coli Bacteria Induces an Increase in the Secretion of Extracellular Vesicle Polydispersity in Macrophages: Possible Involvement of Secreted EVs in the Diagnosis of COVID-19 with Bacterial Coinfections"

_ijms, 2025, doi:10.3390/ijms26083741_

Round 1

Reviewer 1 Report

Comments and Suggestions for Authors

In this article, authors obtained a fraction of bacteria from Escherichia coli, which could be a stimulator for the secretion of a wide variety of EVs that could be monitored through LMW-PTP by confocal microscopy and whose common markers emitted by macrophages can be identified by mass spectrometry. The authors found that patients showed a higher amount of surrounding EVs as they were suffered from bacterial coinfection and SARS-CoV-2, partly due to the bacterial presence. The strategy can be used to study the wide variety of EVs emitted by macrophages, and the EVs related coinfection spread by COVID-19 virus and bacterial. The result is good and deserved to be published. There are some suggestions for authors to consider for the improvement of their article before acceptance.

1.      The abstract looks a little bit of mass. Please make it clearer and more readable. More results of this study are needed in the abstract, not just mentioning some reason to do this study or perspective.

2.     Please give the full name as the non-commonly used abbreviation is first mentioned in the article, even in the abstract, such as EVs.

3.      Their characterization is good, but the analysis is not so clear and comprehensive, and no clear conclusion. Please summarize the detailed mechanism more clearly and descript it more logically on the increased amount of surrounding EVs as the patients are suffered from both virus and bacterial. Are there some synergistic effects between virus and bacterial and how each one involves in this effect and the final result? How is these effects related to tyrosine phosphatases, miRNA and the out membrane proteins?

Comments on the Quality of English Language

 Please do English proof again, such as “bacterial ones”, “sur-rounding”, “Macrophage Large and short EVs”, “Scanning electron microscopy (SEM and TEM)”, etc

Author Response

In this article, authors obtained a fraction of bacteria from Escherichia coli, which could be a stimulator for the secretion of a wide variety of EVs that could be monitored through LMW-PTP by confocal microscopy and whose common markers emitted by macrophages can be identified by mass spectrometry. The authors found that patients showed a higher amount of surrounding EVs as they were suffered from bacterial coinfection and SARS-CoV-2, partly due to the bacterial presence. The strategy can be used to study the wide variety of EVs emitted by macrophages, and the EVs related coinfection spread by COVID-19 virus and bacterial. The result is good and deserved to be published. There are some suggestions for authors to consider for the improvement of their article before acceptance.

 We appreciate the reviewer for their valuable comment and time devoted to our writing.

Comments 1.  The abstract looks a little bit of mass. Please make it clearer and more readable. More results of this study are needed in the abstract, not just mentioning some reason to do this study or perspective.

Response 1. We appreciate the reviewer’s comment. We have added the main results to the abstract.

Comments 2.  Please give the full name as the non-commonly used abbreviation is first mentioned in the article, even in the abstract, such as EVs.

Response 2. We appreciate the reviewer’s comments. We have included the full names of the relevant abbreviations in the paper in the abstract, as well as at the first mentioned of each section of the paper.

  Comments 3.  Their characterization is good, but the analysis is not so clear and comprehensive, and no clear conclusion. Please summarize the detailed mechanism more clearly and descript it more logically on the increased amount of surrounding EVs as the patients are suffered from both virus and bacterial. Are there some synergistic effects between virus and bacterial and how each one involves in this effect and the final result?

Response 3. We appreciate the reviewer for their valuable comment. In results, we have found that the fraction of bacteria evaluate is capable of inducing a significant increase in EVs. In introduction, we have foun that recent studies have shown that SARS-CoV-2 employs small EVs to spread to neighboring cells. Analyzing the background and our results, these suggest that there is a synergistic effect, which is possibly taking advantage of the virus and in turn aggravating the disease. We have also proposed a scheme that summarizes the possible routes that are leading to the production of EVs and the loading of SARS-CoV-2 genes into EVs. Figure 7 Schematic model of polydisperse extracelular vesicles secretion by macrophages in the context of bacterial-SARS-CoV-2 coinfection.

How is these effects related to tyrosine phosphatases, miRNA and the out membrane proteins?

We have identified LMW-PTPs as good markers to monitor the short and large EVs of macrophages that we have produced, using fluorescence confocal microscopy, and this is the purpose that we ultimately gave to the LMW-PTPs in the present work. We consider it a good perspective to carry out further studies focused on evaluating the role played by LMW-PTPs in EVs in various diseases, including bacterial infection and co-infection, as well as their influence on other biomolecules such as RNA.

Comments on the Quality of English Language

 Please do English proof again, such as “bacterial ones”, “sur-rounding”, “Macrophage Large and short EVs”, “Scanning electron microscopy (SEM and TEM)”, etc

We appreciate the reviewer's feedback. We have used the service offered by MDPI to resolve the writing issues.

Reviewer 2 Report

Comments and Suggestions for Authors

I think was obtained a very interesting study model of a variety of EVs that can be produced in vitro. It  could be  important to observe an increase in the amount of EVs in patients with bacterial co-infection and SARS-CoV-2. 

Author Response

I think was obtained a very interesting study model of a variety of EVs that can be produced in vitro. It could be  important to observe an increase in the amount of EVs in patients with bacterial co-infection and SARS-CoV-2. 

We appreciate the reviewer for their valuable comment. We believe that our results provide information that helps to elucidate and better understand part of what has happened in patients who have worsened during infection with SARS-CoV-2, such as the increase in the number of EVs due to bacterial co-infection. Additionally, the strategy of inducing a wide and abundant variety of macrophage EVs in vitro will facilitate their use in different studies.

Reviewer 3 Report

Comments and Suggestions for Authors

I find the manuscript well written in good quality of English. It presents interesting and important topic. COVID-19 had a significant impact on health, social and economic life over the world. All current and future studies will evaluate the different effects on individuals affected by COVID-19. The title is precise and informative and the abstract clearly illustrated the aims of this article. But, in some places it lacks some focus and need refinement.

  1. Introduction

The main topic of this paper is clearly introduced, references are relevant and recent.

  • give the full name of the the abbreviation EV - Extracellular vesicles (EVs)
  • line 90 - "antibiotics have been used for the prevention of bacterial coinfections" may be "used to prevent possible complications"
  • line 94 - add theese Haemophilus influenza and Klebsiella spp.because there is data about them related to severe Covid - 19 ( for example Fan, H., Zhou, L., Lv, J. et al. Bacterial coinfections contribute to severe COVID-19 in winter. Cell Res 33, 562–564 (2023). https://doi.org/10.1038/s41422-023-00821-3).
  • lack focus on whether co-infections are related to the severe Covid-19, to hospitalization, or to Covid-19 patients in general.
  • According to the context in the Introduction, the authors can provide additional information about the role of bacterial extracellular vesicles in the immune response to pathogens as figure/scheme.
  1. Materials and Methods
  • the methods are adequately described, but in too much detail, precise them.
  • move them before results - I think it will be clearer for readers which methods and how were used for the results obtained
  • human samples - all the variables in the text are very well defined but here it is missin the age range, gender or other demographic characteristics. Were all patients hospitalized? Did all have a confirmed positive COVID-19 test?
  • line 547 – “at 30 and 120 min of stimulation” - The sentence is little confusing, please correct. Also, the range between 30 and 120 min is quite large, please explain why there is no intermediate value such as 60 min.

III. Results

The results are clearly presented and illustrated with figures and tables. The practical meaning of them is not clear enough. Statistically significant results are highlighted.

  • line 266 – 270 - what % is the data for Acinetobacter baumannii, because A. baumannii occurs in a large number of Covid-19 patients, especially in a COVID-19 patient of VAP.The COVID-19 pandemic has generated an overuse of antimicrobials in critically ill patients. Acinetobacter baumannii frequently causes nosocomial infections, particularly in intensive care units (ICUs), where the incidence has increased over time.
  1. Discussion

The results are discussed from different angles but again the practical meaning of them is not clear enough.

  • line 381 – “it is expected that more protocols will be established to induce the secretion of EVs in the laboratory” - How do you see these results being integrated into clinical practice? Can you suggest a similar protocol? The methods used are quite laborious, requiring financial and human resources.

Others comments:

  • if study has limitations, they should be written more clearly in a paragraph
  • there is no conclusions that summarizes and guide the reader to the importance of the problem

Overall, the quality of this manuscript is excellent and is very well written. After the refinement, it can be accepted for publication.

Author Response

I find the manuscript well written in good quality of English. It presents interesting and important topic. COVID-19 had a significant impact on health, social and economic life over the world. All current and future studies will evaluate the different effects on individuals affected by COVID-19. The title is precise and informative and the abstract clearly illustrated the aims of this article. But, in some places it lacks some focus and need refinement.

 We appreciate the reviewer for their valuable comment and the time spent reviewing our writing.

  1. Introduction

The main topic of this paper is clearly introduced, references are relevant and recent.

Comments 1. give the full name of the the abbreviation EV - Extracellular vesicles (EVs)

Response 1. We appreciate the reviewer’s comment. We have included the full names of the relevant abbreviations in the abstract, as well as at the first mentioned of each section of the paper.

Comments  2. line 90 - "antibiotics have been used for the prevention of bacterial coinfections" may be "used to prevent possible complications"

Response 2. We thanks the review’s comment. We have used the suggested phrase.

  • Comments 3 line 94 - add theese Haemophilus influenza and Klebsiella spp.because there is data about them related to severe Covid - 19 ( for example Fan, H., Zhou, L., Lv, J. et al. Bacterial coinfections contribute to severe COVID-19 in winter. Cell Res 33, 562–564 (2023). https://doi.org/10.1038/s41422-023-00821-3).

Reponse 3 We appreciate the information provided.  We have used the information found on line 89 – ‘Haemophilus influenzae, and Klebsiella spp. [22]’.

  • Comments 4. lack focus on whether co-infections are related to the severe Covid-19, to hospitalization, or to Covid-19 patients in general.

Response .  We appreciate the reviewer´s insightful comment. In the present study, all patients who were in the condition of moderate or severe COVID-19 disease had bacterial coinfection and were hospitalized. In our study, 85.54% of patients died. More details have been added in section 2.3. Bacterial coinfections analysis in COVID-19 patients.

Comments 5. According to the context in the Introduction, the authors can provide additional information about the role of bacterial extracellular vesicles in the immune response to pathogens as figure/scheme.

Resonse 5. We appreciate the reviewer´s insightful comment. We have found thpt the fraction of bacteria evaluated is capable of inducing a significant increase in EVs. We have found in Introduction that the SARS-CoV-2 virus also use host EVs. Analyzing the evidence and our results, these suggest that there is a synergistic effect, which is possibly taking advantage of the virus and in turn aggravating the disease. We have also proposed a scheme that summarizes the possible routes that are leading to the production of EVs and the loading of SARS-CoV-2 genes into EVs. Some of the receptors that are suggested to be involved in the substrate-macrophage interaction process are TL2, TL4, Scavenger. We added Figure 7 Schematic model of polydisperse extracellular vesicles secretion by macrophages in the context of bacterial-SARS-CoV-2 coinfection

  1. Materials and Methods

Comments 6. the methods are adequately described, but in too much detail, precise them.

Response 6 We thanks the review’s comment. We have addressed the comment, mainly in:

5.3. Cloning, expression, and purification of rHsLMW-PTP

5.10. Liquid chromatography and MALDI MS/MS

Comments 7.  move them before results - I think it will be clearer for readers which methods and how were used for the results obtained

Response 7. We appreciate the review’s comment. We have decided to keep the instructions shown in the ‘Microsoft Word Template‘ provided by MDPI, International Journal of Molecular Sciences, Instructions (https://www.mdpi.com/journal/ijms/instructions)

Comments 8. human samples - all the variables in the text are very well defined but here it is missin the age range, gender or other demographic characteristics. Were all patients hospitalized? Did all have a confirmed positive COVID-19 test?

Response 8. We thank the reviewer for the important observation. The patients were selected according to the following inclusion criteria: a mean age of 57.20 years, with a range of 18 to 60. In total, 33.73% of the patient population were women and 66.26% were men. All patients had moderated to severe disease according to WHO criteria, with an average oxygen saturation of 69% (all patients were hospitalized). In the present study, all patients who were in the condition of moderate or severe COVID-19 disease had bacterial coinfection and were hospitalized. In our study, 85.54% of patients died. More details have been added in section 2.3. Bacterial coinfections analysis in COVID-19 patients.

Male and female patients aged 18 to 60 years with suspected SARS-CoV-2 infection were selected. More details have been added in section 5.1 Human samples for assays

Comments 9. line 547 – “at 30 and 120 min of stimulation” - The sentence is little confusing, please correct. Also, the range between 30 and 120 min is quite large, please explain why there is no intermediate value such as 60 min.

We thanks the review’s comment. We have modified the following sentences.

Response 9. Thirty minutes after induction with SDS-SBMF, some macrophages secreting polydisperse EVs were observed. They were also observed at 60 min and 90 min (data not shown), reaching the optimal time at 120 min.

Figure 2. Large and short EVs were induced by the bacterial fraction of E. coli and recognized by anti-LMW-PTPs. Macrophages were stimulated for 30 still 120 min at 37 ºC with bacterial extraction (SDS-SBMF), or unstimulated (cells that spontaneously released EVs were sought, they are <20%).

III. Results

The results are clearly presented and illustrated with figures and tables. The practical meaning of them is not clear enough. Statistically significant results are highlighted.

We appreciate the review’s comment. We have attended to the description of all the figures.

Comments 10. line 266 – 270 - what % is the data for Acinetobacter baumannii, because A. baumannii occurs in a large number of Covid-19 patients, especially in a COVID-19 patient of VAP.The COVID-19 pandemic has generated an overuse of antimicrobials in critically ill patients. Acinetobacter baumannii frequently causes nosocomial infections, particularly in intensive care units (ICUs), where the incidence has increased over time.

Response 10. We thank the reviewer for the important observation. We have added the description at the end of the section: 23. Bacterial co-infections analysis in COVID-19 patients. Acinetobacter baumannii was present in only n=3 patients because the samples were at the beginning of the hospital stay taken.

Therefore, the patients had not yet been exposed to hospital infections.

  1. Discussion

The results are discussed from different angles but again the practical meaning of them is not clear enough.

Comments 11. line 381 – “it is expected that more protocols will be established to induce the secretion of EVs in the laboratory” - How do you see these results being integrated into clinical practice? Can you suggest a similar protocol? The methods used are quite laborious, requiring financial and human resources.

Response 11 Therefore, the patients had not yet been exposed to hospital infections.

- Our results suggest that the strategy we used to obtain the variety of small to large EVs could be considered for future work, seeking to better adapt the strategy to insolation, mass spectrometry and other analyses.

- Our working group will evaluate applications in different cell types, simulating the secretion of polydisperse EVs to analyze diseases.

- The method we have generated provides a relatively simple way to produce abundant amounts of macrophage EVs in a short time, requiring common equipment in laboratories where cell cultures are performed.

Others comments:

Comments. 12 if study has limitations, they should be written more clearly in a paragraph

We appreciate review’s comment.

Response 12.  Our model of induction of EV secretion in macrophages has the limitation that it requires being adapted to the cell line in which the effect is to be evaluated, first analyzing whether this strategy works, and we have placed it in the Discussion.

  • Comments 13. there is no conclusions that summarizes and guide the reader to the importance of the problem

We thank the reviewer for the important observation. We have placed the Conclusions in a section:

Response 12. 4. Conclusions. We have developed a method to induce macrophages to secrete a wide variety of polydisperse populations of extracellular vesicles in vitro through the bacterial fraction of E. coli. A variety of populations could be present in bacterial coinfections with SARS-CoV-2, which we believe has participated in aggravating clinical manifestations in patients, increasing the spread of the virus, as 85.5% of the patients in this study died. Our in vitro method of inducing extracellular vesicles could be used to obtain larger samples for study and the detection of diagnostic and prognostic biomarkers of different diseases.Overall, the quality of this manuscript is excellent and is very well written. After the refinement, it can be accepted for publication.

Overall, the quality of this manuscript is excellent and is very well written. After the refinement, it can be accepted for publication.

Round 2

Reviewer 1 Report

Comments and Suggestions for Authors

Now the article is well-written and I am satisfied with it.

Author Response

(The authors gave the same response as above.)

Reviewer 3 Report

Comments and Suggestions for Authors

Dear Authors,
I have been carefully reviewed your revised manuscript with the ID number “ijms-3390920”. In my opinion, this revised article includes all the points raised in the original draft. 
Best wishes to all of the authors who contributed this wonderful work.

Author Response

Reviewer 3

I find the manuscript well written in good quality of English. It presents interesting and important topic. COVID-19 had a significant impact on health, social and economic life over the world. All current and future studies will evaluate the different effects on individuals affected by COVID-19. The title is precise and informative and the abstract clearly illustrated the aims of this article. But, in some places it lacks some focus and need refinement.

 We appreciate the reviewer for their valuable comment and the time spent reviewing our writing.

  1. Introduction

The main topic of this paper is clearly introduced, references are relevant and recent.

Comments 1. give the full name of the the abbreviation EV - Extracellular vesicles (EVs)

Response 1. We appreciate the reviewer’s comment. We have included the full names of the relevant abbreviations in the abstract, as well as at the first mentioned of each section of the paper.

Comments  2. line 90 - "antibiotics have been used for the prevention of bacterial coinfections" may be "used to prevent possible complications"

Response 2. We thanks the review’s comment. We have used the suggested phrase.

  • Comments 3 line 94 - add theese Haemophilus influenza and Klebsiella spp.because there is data about them related to severe Covid - 19 ( for example Fan, H., Zhou, L., Lv, J. et al. Bacterial coinfections contribute to severe COVID-19 in winter. Cell Res 33, 562–564 (2023). https://doi.org/10.1038/s41422-023-00821-3).

Reponse 3 We appreciate the information provided.  We have used the information found on line 89 – ‘Haemophilus influenzae, and Klebsiella spp. [22]’.

  • Comments 4. lack focus on whether co-infections are related to the severe Covid-19, to hospitalization, or to Covid-19 patients in general.

Response .  We appreciate the reviewer´s insightful comment. In the present study, all patients who were in the condition of moderate or severe COVID-19 disease had bacterial coinfection and were hospitalized. In our study, 85.54% of patients died. More details have been added in section 2.3. Bacterial coinfections analysis in COVID-19 patients.

Comments 5. According to the context in the Introduction, the authors can provide additional information about the role of bacterial extracellular vesicles in the immune response to pathogens as figure/scheme.

Resonse 5. We appreciate the reviewer´s insightful comment. We have found thpt the fraction of bacteria evaluated is capable of inducing a significant increase in EVs. We have found in Introduction that the SARS-CoV-2 virus also use host EVs. Analyzing the evidence and our results, these suggest that there is a synergistic effect, which is possibly taking advantage of the virus and in turn aggravating the disease. We have also proposed a scheme that summarizes the possible routes that are leading to the production of EVs and the loading of SARS-CoV-2 genes into EVs. Some of the receptors that are suggested to be involved in the substrate-macrophage interaction process are TL2, TL4, Scavenger. We added Figure 7 Schematic model of polydisperse extracellular vesicles secretion by macrophages in the context of bacterial-SARS-CoV-2 coinfection

  1. Materials and Methods

Comments 6. the methods are adequately described, but in too much detail, precise them.

Response 6 We thanks the review’s comment. We have addressed the comment, mainly in:

5.3. Cloning, expression, and purification of rHsLMW-PTP

5.10. Liquid chromatography and MALDI MS/MS

Comments 7.  move them before results - I think it will be clearer for readers which methods and how were used for the results obtained

Response 7. We appreciate the review’s comment. We have decided to keep the instructions shown in the ‘Microsoft Word Template‘ provided by MDPI, International Journal of Molecular Sciences, Instructions (https://www.mdpi.com/journal/ijms/instructions)

Comments 8. human samples - all the variables in the text are very well defined but here it is missin the age range, gender or other demographic characteristics. Were all patients hospitalized? Did all have a confirmed positive COVID-19 test?

Response 8. We thank the reviewer for the important observation. The patients were selected according to the following inclusion criteria: a mean age of 57.20 years, with a range of 18 to 60. In total, 33.73% of the patient population were women and 66.26% were men. All patients had moderated to severe disease according to WHO criteria, with an average oxygen saturation of 69% (all patients were hospitalized). In the present study, all patients who were in the condition of moderate or severe COVID-19 disease had bacterial coinfection and were hospitalized. In our study, 85.54% of patients died. More details have been added in section 2.3. Bacterial coinfections analysis in COVID-19 patients.

Male and female patients aged 18 to 60 years with suspected SARS-CoV-2 infection were selected. More details have been added in section 5.1 Human samples for assays

Comments 9. line 547 – “at 30 and 120 min of stimulation” - The sentence is little confusing, please correct. Also, the range between 30 and 120 min is quite large, please explain why there is no intermediate value such as 60 min.

We thanks the review’s comment. We have modified the following sentences.

Response 9. Thirty minutes after induction with SDS-SBMF, some macrophages secreting polydisperse EVs were observed. They were also observed at 60 min and 90 min (data not shown), reaching the optimal time at 120 min.

Figure 2. Large and short EVs were induced by the bacterial fraction of E. coli and recognized by anti-LMW-PTPs. Macrophages were stimulated for 30 still 120 min at 37 ºC with bacterial extraction (SDS-SBMF), or unstimulated (cells that spontaneously released EVs were sought, they are <20%).

III. Results

The results are clearly presented and illustrated with figures and tables. The practical meaning of them is not clear enough. Statistically significant results are highlighted.

= We appreciate the review’s comment. We have attended to the description of all the figures.

Comments 10. line 266 – 270 - what % is the data for Acinetobacter baumannii, because A. baumannii occurs in a large number of Covid-19 patients, especially in a COVID-19 patient of VAP.The COVID-19 pandemic has generated an overuse of antimicrobials in critically ill patients. Acinetobacter baumannii frequently causes nosocomial infections, particularly in intensive care units (ICUs), where the incidence has increased over time.

Response 10. We thank the reviewer for the important observation. We have added the description at the end of the section: 23. Bacterial co-infections analysis in COVID-19 patients. Acinetobacter baumannii was present in only n=3 patients because the samples were at the beginning of the hospital stay taken.

Therefore, the patients had not yet been exposed to hospital infections.

  1. Discussion

The results are discussed from different angles but again the practical meaning of them is not clear enough.

Comments 11. line 381 – “it is expected that more protocols will be established to induce the secretion of EVs in the laboratory” - How do you see these results being integrated into clinical practice? Can you suggest a similar protocol? The methods used are quite laborious, requiring financial and human resources.

Response 11 Therefore, the patients had not yet been exposed to hospital infections.

- Our results suggest that the strategy we used to obtain the variety of small to large EVs could be considered for future work, seeking to better adapt the strategy to insolation, mass spectrometry and other analyses.

- Our working group will evaluate applications in different cell types, simulating the secretion of polydisperse EVs to analyze diseases.

- The method we have generated provides a relatively simple way to produce abundant amounts of macrophage EVs in a short time, requiring common equipment in laboratories where cell cultures are performed.

Others comments:

Comments. 12 if study has limitations, they should be written more clearly in a paragraph

We appreciate review’s comment.

Response 12.  Our model of induction of EV secretion in macrophages has the limitation that it requires being adapted to the cell line in which the effect is to be evaluated, first analyzing whether this strategy works, and we have placed it in the Discussion.

  • Comments 13. there is no conclusions that summarizes and guide the reader to the importance of the problem

We thank the reviewer for the important observation. We have placed the Conclusions in a section:

Response 12. 4. Conclusions. We have developed a method to induce macrophages to secrete a wide variety of polydisperse populations of extracellular vesicles in vitro through the bacterial fraction of E. coli. A variety of populations could be present in bacterial coinfections with SARS-CoV-2, which we believe has participated in aggravating clinical manifestations in patients, increasing the spread of the virus, as 85.5% of the patients in this study died. Our in vitro method of inducing extracellular vesicles could be used to obtain larger samples for study and the detection of diagnostic and prognostic biomarkers of different diseases.Overall, the quality of this manuscript is excellent and is very well written. After the refinement, it can be accepted for publication.

Overall, the quality of this manuscript is excellent and is very well written. After the refinement, it can be accepted for publication.
